# Landscape-Based Visions as Powerful Boundary Objects in Spatial Planning: Lessons from Three Dutch Projects

**Sabine van Rooij** [1], **Wim Timmermans** [1], **Onno Roosenschoon** [1], **Saskia Keesstra** [1,2], **Marjolein Sterk** [1] and **Bas Pedroli** [3,*]

1   Wageningen Environmental Research, P.O. Box 47 6700 AA Wageningen, The Netherlands;
    sabine.vanrooij@wur.nl (S.v.R.); wim.timmermans@wur.nl (W.T.); onno.roosenschoon@wur.nl (O.R.);
    saskia.keesstra@wur.nl (S.K.); marjolein.sterk@wur.nl (M.S.)
2   Civil, Surveying and Environmental Engineering, The University of Newcastle, Callaghan 2308, Australia
3   Land Use Planning Group, Wageningen University, P.O. Box 47 6700 AA Wageningen, The Netherlands
*   Correspondence: bas.pedroli@wur.nl; Tel.: +31-317-485-396

**Abstract:** In a context of a rapidly changing livability of towns and countryside, climate change and biodiversity decrease, this paper introduces a landscape-based planning approach to regional spatial policy challenges allowing a regime shift towards a future land system resilient to external pressures. The concept of nature-based solutions and transition theory are combined in this approach, in which co-created normative future visions serve as boundary concepts. Rather than as an object in itself, the landscape is considered as a comprehensive principle, to which all spatial processes are inherently related. We illustrate this approach with three projects in the Netherlands in which landscape-based visions were used to guide the land transition, going beyond the traditional nature-based solutions. The projects studied show that a shared long-term future landscape vision is a powerful boundary concept and a crucial source of inspiration for a coherent design approach to solve today's spatial planning problems. Further, they show that cherishing abiotic differences in the landscape enhances sustainable and resilient landscapes, that co-creation in the social network is a prerequisite for shared solutions, and that a landscape-based approach enhances future-proof land-use transitions to adaptive, circular, and biodiverse landscapes.

**Keywords:** nature-based solutions; transition; regional planning; landscape management; future vision; circularity; resource management; biodiversity

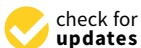



## 1. Introduction

The livability of the city and the countryside is under great pressure all over the world. Cities face major challenges, such as expanding housing development, densification, flood prevention, and biodiversity decline [1]. In rural areas, waterlogging and drought are increasingly uncertain factors for agriculture. Drought is also a problem for nature areas, with nitrogen deposition recently labeled an acute additional threat to biodiversity [2,3]. At the same time, there is a growing awareness among governments, citizens and the business community that the planning tasks facing these challenges cannot be realized without a coherent vision of the future of our landscape [4]. In addition to emergency measures for the short term, well-thought-out long-term adjustments to changing conditions are necessary. Reference is made here to transition theory [5]: once a real transition is required, a regime shift should take place. In a spatial context, the transition should be based on the landscape as a vehicle for spatial planning since the landscape provides both the physical and the perceived baseline for spatial development [6]. Given the complex character of the current landscape planning issues, climate-robust biodiversity and circularity are key principles for responsible landscape adaptation in such spatial processes [7]. In this context, it is crucial that all actors take part in the spatial planning process and that they

can contribute their local knowledge and opinions so that the emerging strategies are not only tailored to the biophysical, but also to the social landscape [8].

Finding solutions for this ill-defined Gordian knot of challenges spatial planning is facing means that actions of different sectors and actors need to be aligned. When each sector or actor approaches the challenges as a well-structured sectoral problem, there is little chance that visions and actions will come together in an adequate strategy for a region [4,6]. Another approach is to acknowledge that the solution for the knot of challenges is a boundary concept. A solution for these challenges is not part of our present-day infrastructure, our conventional social arrangements and technologies [9]. A future vision for a region, based on a set of conceptual principles such as "landscape-based" can serve as a boundary concept, offering common ground to the scientists and practitioners with different backgrounds, values and interests that were involved [6]. A boundary concept is flexible enough to adapt to local needs and to different perspectives, but also robust enough to maintain conceptual coherence across scientific disciplines and across the science-practice boundary [9,10]. Planning literature [11] reflects this shift from single fixed quantitative targets, via multiple, qualitative concepts to the guidance of interactions in a multi-stakeholder perspective and finally even fuzzy planning approaches.

Nature-based solutions are spatial interventions that use natural materials and especially natural principles. Nature-based solutions can be considered an umbrella term for all related applications of ecosystem services, natural capital and "lessons from nature" [12]. For governments and companies (construction, infrastructure), these types of solutions are an encouraging reference because multiple goals can be combined in one measure. When applying nature-based solutions, the emphasis is often on solving problems in urban and rural planning by making use of the processes and patterns of nature [13]. So far, solutions are often sought within the scale level of the project area. For adaptation to changing circumstances in urban and rural areas, it is important to search for solutions on a broader scale level, the landscape level, because the city and the countryside are interconnected systems [6]. Nature-based solutions are ideally suited to link those scales and to use a systems approach that uses the natural processes instead of working against them.

In order to strive for an integral solution in a spatial context and incorporating the social environment [14], this paper introduces a landscape-based planning approach aiming at regional spatial policies allowing for a community-based transition in a relatively urbanized countryside resilient to various external pressures. Three examples from the Netherlands serve as an illustration of three topical transition issues in a metropolitan context: climate adaptation, biodiversity enhancement and flood risk. Questions to be answered are how a landscape-based approach adds to nature-based solutions, how the abiotic landscape can be considered in terms of opportunities, how future visions can support sound transitions and how local and regional planning can reinforce each other.

## 2. Landscape: A Concept rather than an Object

### 2.1. Multiplicity of Landscape

Landscape according to the European Landscape Convention is an area, as perceived by people, whose character is the result of the action and interaction of natural and/or human factors [15]. In this paper, we consider landscape as a vehicle for spatial planning, rather than as an object for planning itself. Starting from the basic abiotic differentiation underlying all landscape processes, a landscape-based approach to spatial planning should make use of the opportunities offered by the landscapes further differentiated by societal expectations and cultural norms, instead of designing the landscape according to the economic ambitions of today's users only as, at the end of the day, is still often the dominant practice [16–18].

### 2.2. Urban and Rural Relationships

Soil, topography, water and historical patterns define the opportunities of spatial planning to a large extent. Neglecting these patterns and the associated processes through

intensification and scale-enlargement of land use has led to numerous examples of tragic degradation [19]. Although the countryside is often considered as the rural opposite of the city, both have always been tightly connected [20,21]. The first cities were directly related to the provision of sufficient food, water and energy in the surrounding land and their defense depended on their situation in the landscape. Industrialization favored the growth of cities connected to the availability of resources such as coal and metal and to transport networks of rivers, over seas and over land. Brenner [22] describes the next step of planetary urbanization with cities worldwide better connected to each other than to their surrounding landscape. Timmermans, Woestenburg, Annema, Jonkhof, Shlakku and Yano [21] consider European capitals as cities that, through centuries, have successfully managed to profit from the different chances offered by their surrounding landscapes, while other cities faced degeneration when the landscape did not offer enough of what they needed for further growth.

### 2.3. Land use Transitions

Landscape as a concept is a crucial element in land-use transitions [23]. Transition science addresses the interplay between humans and the systems around them in which they operate. The key characteristic is that it is oriented on system and policy innovation. This is essential, as business as usual, or even innovations that optimize the current situation, are insufficient to resolve the issues of our time [24]. As a key contribution to transition science, Rotmans describes a new world view that can help to define actions to facilitate the now urgently needed transitions in our society [5,25]. In this world view, problems are solved in a cooperation model (as opposed to an exploitation model), in which business models are not focused on economic return, but on societal return and value creation instead of value extraction. Key to this framework is the transition cross (X-curve) of Visser, Keesstra, Maas and de Cleen [8] in which they explain the process from the old system towards a new system. In the X-curve, transition is described as a process of construction and demolition, which usually has a long pre-development phase (decades) and the real transition phase is relatively short (years), and characterized by chaotic and disruptive events (compare the adaptive cycle in ecosystems of Holling [26]). In Figure 1, this conceptual model is further elaborated for transitions in landscape adaptation processes. When we follow the green line, the lower left part represents the start of innovative new approaches. They start as a niche product that is developed in an "Experiment" phase. Once the approach or product has proven to be useful, it gains popularity in the "Acceleration" and "Emergence" phases, but still is seen as a niche product. To get to the other side of the chaotic transformation phase, enablers are needed to shift to the "Institutionalization" and "Stabilization" phases in which the approach or product becomes the new normal. Apart from this positive transition towards more sustainable approaches and products, it is also important to have attention for the phasing out of unsustainable approaches and products, which is depicted by the grey line. Many innovations are not truly transformative, and only optimize business as usual. The phasing out needs to incorporate the dependencies and lock-ins that form barriers for change. This process has several steps: "corrective barriers", "reduce dependencies" and "reduce relevance" to reach the phasing out.

This framework aims to enable changes and actions that should be taken to support sustainability in the short- and long-term and give direction to necessary actions in land restoration, sustainable land use and management and land and soil policy. The framework can provide the required intensive guidance to (i) analyze the impact of incentives, (ii) identify new reference points in the transition and (iii) stimulate transition catalysts, and (iv) innovate by testing cutting edge policy instruments in close cooperation with society [8].

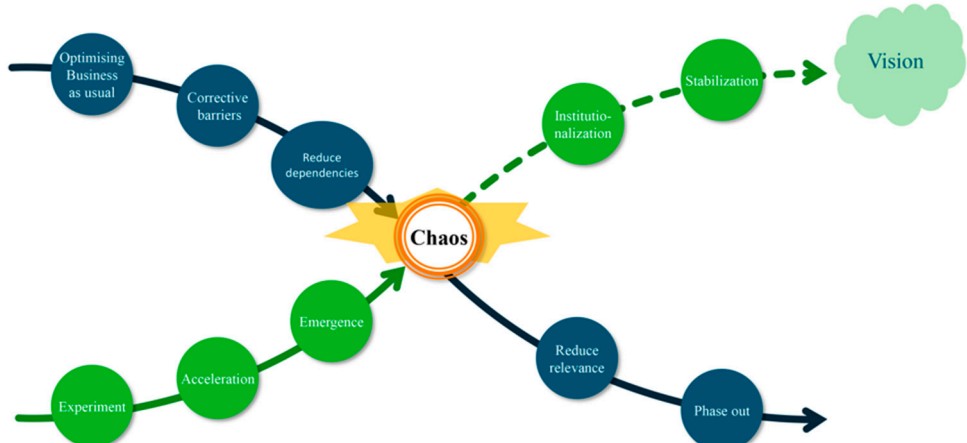

**Figure 1.** The X-curve transition cross (adapted after Visser, Keesstra, Maas and de Cleen [8]).

## 3. Characteristics of Landscape-Based Planning

Much has been written about nature-based solutions. In this paper, we do consider nature-based solutions a guiding principle for solving the knot of challenges mentioned in the introduction that society is facing today. Still, we prefer to slightly reframe the concept:

- using the term "landscape-based planning" rather than nature-based solutions, as it combines the social and environmental systems, instead of focussing on ecosystem processes. The concept will identify from the start of the planning process the opportunities that the landscape dimension (the spatial-temporal dimension including biophysical, social and cultural aspects) can offer [27].
- using "landscape-based" rather than "nature-based": Herewith we follow the reasoning of Termorshuizen and Opdam [28] that the concept of landscape services in metropolitan Europe is more appropriate than the concept of ecosystem services. They chose the concept of landscape services over the concept of ecosystem services as it better associates with pattern–process relationships, it better unifies scientific disciplines and it is more relevant and legitimate to local practitioners. People live in landscapes, not in ecosystems. It informs the actors with sound knowledge about how to best reconcile their needs to the landscape structure and processes.
- using "landscape-based planning" rather than "landscape-based solutions". In northwest Europe, one can observe a strong tradition of strict land-use planning on our scarcely available land. We have a history of dominating strong natural processes like flooding instead of letting land-use be the consequence of these processes. Further, a rich knowledge basis has been developed about the conditions under which landscape services can emerge [29]. This knowledge is a vital key to restore the landscape services that are needed as a solution for spatial planning challenges. This requires well-thought-out land-use planning, principally fostering a co-creating practice to fully account for the societal aspects of future developments. Focus is not so much on solutions for individual problems in the spatial development, but rather on a comprehensive planning approach encompassing as much as possible the potentially emerging problems in the future.

The next section describes some examples of a landscape-based approach, highlighting the guidance steps as defined in the previous section.

## 4. Examples of Landscape-Based Approaches

### 4.1. Example A: Regional Adaptation Strategy for the Region Vallei and Veluwe

#### 4.1.1. Reflection on Incentives

Climate change is happening, and it is forecasted that we will face more extreme weather events, more often. Therefore, in the Netherlands, a National Adaptation Policy

was put into place. As a follow-up to that, in the Vallei and Veluwe region (2456 km$^2$), a regional coalition of 28 municipalities, the water board, two provinces, the drinking water company and the regional security board co-created a Regional Adaptation Strategy for this strategy, a landscape-based approach was adopted. This approach is analogous to a co-production principle for landscape governance and transformation [29,30] and underpins a new governance style of local urban climate adaptation [31].

The Veluwe is a hilly mosaic landscape with dry sandy areas where the villages were traditionally located and low-lying areas with shallow groundwater tables with new neighborhoods where heavy rainfall causes floods and droughts are frequent in dry periods. For the area, a "climate effect atlas" was developed, including information about the long-term effects of floods and droughts [32].

### 4.1.2. Definition of New Reference Points for Transition

To help the process of transition to a situation where water management is adapted to the changing climate, a so-called "climate effect atlas" was produced, containing important reference points for adaptation. The climate effect atlas was produced in three steps: (1) data from the current climate and the expected climate in 2050 were retrieved from the Dutch national weather service (Royal Netherlands Meteorological Institute, KNMI); (2) a hydrological model was made, using the climate data and current landscape data, with which maps with the frequency of flood events now and in the future could be generated; (3) a map was produced that reflects the hydrological functioning of the natural system of the region, based on the combination of soil and geomorphological data.

In the Regional Adaptation Strategy, the current system and climate effect atlas were compared, which showed that the current natural and water management systems are unable to adequately regulate the water volumes associated with these excessive rains and prolonged droughts.

The climate effect atlas gives regional actors more insight into the functioning and the limits of the natural system regarding run-off, the impact of climate change on this system and on different types of land use. These insights were used to set up a Regional Adaptation Strategy and will further find their way in a Regional Adaptation Plan, that will be developed as a follow-up.

### 4.1.3. Identification of Transition Catalysts

The main catalyst of the Vallei and Veluwe approach was the awareness of upcoming climate change. This resulted in a National Adaptation Policy that is now regionally implemented. The catalyzing factor that led to choosing an innovative attitude, where technology and engineering follow natural processes in the system instead of working against them was a visionary group of policymakers and politicians who succeeded in convincing the stakeholders in the region to adopt a systems-, or landscape-based approach. Interestingly, this might be considered the refurbishing of the Design with Nature concept of McHarg [33], in this case emphasizing a participative stakeholder approach.

### 4.1.4. Innovation by Testing Policy Instruments

In the next steps, the science-base will be introduced to the Regional Adaptation Plan, defining a new policy for dealing with drought and flood events and new engineering standards. Administrative support is now in place for a landscape-based approach, integrating traditional man-made infrastructures such as technical sewage systems and water bodies, with landscape-based measures and upcoming nature-based techniques such as community-based initiatives of rainwater filtration and green roofs. A reflective participation approach [34] will be used to test the policy instruments defined.

*4.2. Example B: Regional Case of Bee Landscapes: A Socio-Ecological Network for Pollinators*

4.2.1. Reflection on Incentives

Wild pollinators have drastically declined in occurrence and diversity at local and regional scales in North-West Europe during the last decades [17,35]. Land-use change, and the current land management intensity are the most important causes [17]. Greater landscape-scale habitat diversity often results in more diverse pollinator communities and more effective crop and wild plant pollination. Semi-natural habitats, habitat corridors, landscape heterogeneity and landscape configuration are propagated as incentives to mitigate the negative effects of intensive land use on wild pollinators [35,36].

4.2.2. Definition of New Reference Points for Transition

As a response to these incentives, a regional authority (Province of South Holland), a company (Heineken) and a research institute (Wageningen University and Research) joined forces in 2013 in the "Green Circles" program, and decided to create a "Bee-Landscape" in the region around Leiden, Zoetermeer and Alphen aan de Rijn, as much as possible involving local stakeholders [37,38]. The aim was to initiate and stimulate a transition towards a more sustainable region in the province of South-Holland. The intensive, hands-on and parity-based cooperation between these different parties provides a very new reference point for transition. Using the concept of socio-ecological networks, a social network was set up to stimulate and enable coordinated action to realize an ecologically functional Bee Landscape.

In the initial phase of this initiative, the knowledge institute was asked to give scientifically sound substance to the term "Bee Landscape". The scientific knowledge was communicated in an attractive and easy to understand manner to local stakeholders. In 2016, the growing group of stakeholders that were involved in the Bee Landscape drew up and signed a covenant, in which measurable ambitions were described for the Bee Landscape.

4.2.3. Identification of Transition Catalysts

Green Circles launched the ambition to create a "Bee Landscape" and involved local and regional partners to do so, without specifying what a bee landscape exactly would be. The term "Bee Landscape" served as a boundary concept. The network of actors could lean on the sound knowledge of research institutes involved such as the preconditions for a high diversity of wild pollinators in the landscape and the strategic areas for the improvement of the habitat network. A so-called "Bee Landscape helpdesk" provided the opportunity for actors to invite pollinator specialists to their property for free advice. Further, a monitoring scheme for wild pollinators was put into place, to monitor progress, which was actively shared with the actors in the network.

The network grew from a dozen organizations in 2014 to over 30 in 2020. In 2016, a shared vision on a sustainable Bee Landscape and an associated covenant was drawn up and signed by 20 organizations. In this covenant, the time horizon taken was vague, but far away ("for the future"). This was in line with the spirit of the network: organizations working together and inspiring and helping each other, rather than cooperation based on control and accountability. Also, the use of the term Bee Landscape turned out to be helpful as it served as a strong boundary concept. It helped to move both the social and the bee network forward, even though not all actors had the same image of a comprehensive bee landscape. Last but not least, the participation of leading companies such as the Heineken brewery and AKZO, helped to motivate other parties to join the network.

4.2.4. Innovation by Testing Policy Instruments

The province chose to invest in a network coordinator to set up the network, support the exchange of knowledge and experiences so that the network became a true learning network, supporting the tailoring of scientific knowledge to the needs of the network and in providing a monitoring scheme for pollinators. In addition, they co-financed measures for

wild pollinators in the field. This approach of investing in the emergence of self-managing networks that are committed to solving socio-ecological problems is quite new in the Netherlands and now several other initiatives have used the same approach.

*4.3. Example C: Regional Case—The Plan Ooievaar (Plan "Stork")*

4.3.1. Reflection on Incentives

The Dutch riverine area with the Rhine and Meuse rivers is centrally located in the Netherlands. Due to normalization and embankments, the riverbed between the dikes is now higher in the landscape caused by sedimentation, while the surrounding area has become lower, due to soil subsidence. The resulting river landscape was used almost only for intensive cattle grazing. This all has led to a situation where the high river discharges became a threat for the surrounding area, the biodiversity decreased, and the spatial quality of the area (i.e., the characteristic functional coherence of patterns and processes in the landscape, after [39]) was low.

Therefore, in 1985, a competition was held to find the best design for the landscape development of the riverine area at a regional level. The Plan "Ooievaar" [40], which won the competition, contained a number of new, appealing principles in terms of managing rivers, nature, agriculture and extraction of minerals (clay, sand or gravel). The plan advocated new interactions between the natural dynamics of a river system, the resulting visual expression and spatial quality, and land use. As a result, in the river landscape, agriculture and nature development would go hand in hand by making full use of the agricultural system and ecosystem potentials.

Plan "Ooievaar" promoted interweaving river management, nature development and landscape architecture, it was followed up by several experiments. When in 2001 the Deputy Minister separately commissioned Rijkswaterstaat and Regional governments to develop the National Strategy Room for the River, both parties decided to develop it together, based on positive previous experiences of collaboration [41].

4.3.2. Definition of New Reference Points for Transition

In this plan, the entire Dutch river area was an object of design. This scale level was innovative at that time. Another essential element in this plan is the combination of what is constructed and what unfolds naturally. The man-made part is drawn, described and calculated. The part that develops naturally is a matter of speculation, which does not however mean it happens by chance. The very opposite is true: it is fed by expertise. However, this self-same expertise teaches that the process is a game in which uncertainty and surprise are influencing the outcome.

4.3.3. Identification of Transition Catalysts

"Ooievaar" is the Dutch word for "Stork", and the makers of this plan, relying on their ecological expertise, expected the interventions suggested in the plan to lead to new nature values and an increase in biodiversity. This newly created natural environment would appeal to the black stork, a species characteristic of highly varied river ecosystems, which had left the Netherlands a long time ago. The label "Stork" can be considered as a boundary concept, enabling different sectors to work together. The main transition catalysts were the Non-Governmental Organizations: (i) growing societal resistance to dike reinforcement and (ii) a growing belief that these measures alone could not deliver future flood safety [42]. Then, in 1993 and 1995, two large flood events occurred which showed the vulnerability of our riverine area. The confrontation with the acute flood risk, combined with the first awareness of the effects of climate change on river discharges, contributed to a paradigm shift in flood management towards accommodating floods in a co-creative process.

4.3.4. Innovation by Testing Policy Instruments

In the early 2000s, the National Room for the River Program was launched to increase flood safety by giving the rivers literally more room, combined with increased spatial

quality of landscape, nature and culture [43,44]. This program changed the topography and the water regime of the Dutch rivers profoundly, giving the river dynamics free rein, provoking a chain of transformations that increases biodiversity. Since a new landscape arises—with the black stork as a bonus. The policy instruments used were setting civil engineering boundary conditions for flood safety and navigation and co-creation of nature rehabilitation and landscape plans within this framework.

## 5. Discussion

### 5.1. Lessons Learned from the Examples

Comparing the three examples, common characteristics of the transition pathway become clear, as illustrated in the transition model of Figure 1. Incentives in the field of water management, climate change and biodiversity put pressure on the business as usual. In our highly organized and specialized society, these incentives can affect a wide range of actors. An adequate reaction to these incentives therefore involves a multitude of sectors that are closely interrelated. This requires a transformation of the business as usual. In the examples, a landscape-based approach is used as a guideline for this transition (Figure 2). Starting from the undefined need for change ("chaos"), the abiotic landscape system defines the safe operating space for sustainable development in which a future vision should fit. Visions that are unsustainable in the long term will appear to be unfeasible in an early stage. Shared visions are comprised of a multidimensional target space. In the first phase, pathways should be defined to arrive within the limits of this space, varying under the influence of external factors, such as climate change and societal changes. In time, the boundary concept takes shape and increasingly inspires stakeholders and policymakers, constituting a second phase. At the end of the day, a normative design represents a more narrowly defined point on the planning horizon, that is, a reasonably explicit description of the future landscape. In fact, this visioning of a normative design implies a back-casting approach, implying that the safe operating space is not the only norm. This process will need to be repeated each time new pressing issues appear to become incentives (cfr. adaptive planning [45]).

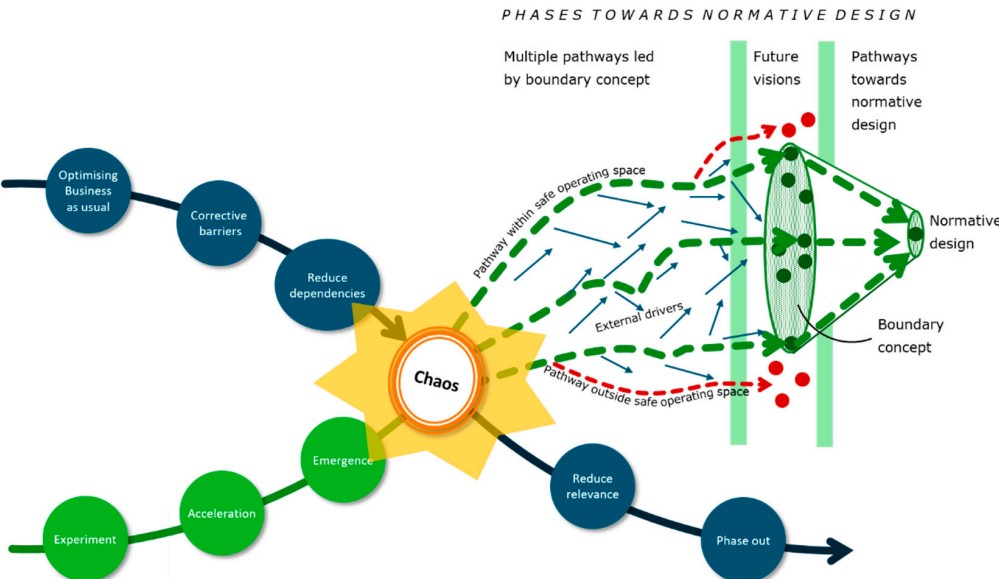

**Figure 2.** The role of visions as boundary concepts. Several sustainable visions (dark green spots) are comprised of multidimensional target space, whereas unsustainable visions (red dots) are unfeasible. Pathways should be defined—and followed—to arrive within the limits of this space, varying under the influence of external factors, such as climate change and societal changes.

The examples can be compared on the basis of the characteristics described in Section 3 (see Table 1).

**Table 1.** Comparison of the examples.

| Characteristics of Land-Use Plan | | Example A:<br>Vallei and Veluwe | Example B:<br>Bee Landscape | Example C:<br>Plan Ooievaar |
|---|---|---|---|---|
| Goal: | | Climate adaptation preventing flooding and droughts | Stimulate population development of pollinating insects | Decrease regional flood risk while stimulating biodiversity and spatial landscape quality |
| Boundary concept adopted: | | Natural landscape system | Bee landscape | Black stork |
| Landscape dimension | Spatial dimensions defined by: | Water dynamics (watershed level) | Population dynamics (landscape level) | Water dynamics (regional/national level) |
| | Time horizon: | Medium to long term | Long term | Medium to long term |
| Landscape services | Biodiversity: | Positive effects of nature-based solutions | More diverse pollinator communities | Newly created natural environment increases biodiversity |
| | Circularity: | Circular practices in wastewater and solid waste management | Pollination and honey production as a win-win situation | Extraction of river sediment for construction industry while alleviating flood risk |
| | Adaptation to climate change: | Flood prevention by many measures; several water-retention solutions; counteracting urban heat island effect | Climate robust habitats for high diversity of pollinators | Adaptation to changing river discharges and extreme flood and drought events |
| Comprehensive planning | Incentives: | Threat of extreme weather events | Decline of pollinators | Increasing flood risk along the rivers |
| | New Reference points for transition: | 2050 climate scenario in map | Parity based hands-on transition management | DNA of the River (basic landscape principles to be taken into consideration) |
| | Transition catalysts: | Waterboard as responsible for climate regional adaptation plan | Diverse network of actors; capable network coordinator | Non-governmental organizations and later sectoral stakeholders |
| | Innovative policy instruments: | Adopting a systems approach to overcome local–regional dichotomy | Financial support for building and running a learning network | Clear boundary conditions for co-creation of nature and landscape rehabilitation |
| Landscape-based approach | | Build on the characteristics of the physical landscape to adapt to changing precipitation patterns | Build on the motivation and willingness in the social landscape system to reverse the decline of pollinators in their landscape | Build on the original natural characteristics of river sections on a national and regional level, to accommodate higher discharges |

The steps and requirements that we distill from the example cases are the following:

- Boundary concept. The introduction of a boundary concept is often summarized in a catchy term and can help actors to agree on goals on an abstract level. This enhances constructive conversations and cooperation. The boundary concept in our examples is the underlying landscape vision, where a catchy term is often the label to which the vision is attached.
- Landscape dimension. Actors need to realize that the landscape as an underlying system is crucial in solving the challenges that they face both in space and in time.

- Landscape services, especially biodiversity, circularity and climate adaptation. Understanding the natural and societal system is a first step in the planning process so that it becomes clear how the landscape can provide the landscape services that people need and to assess the physical and societal capacity of the system to offer opportunities for new arrangements of functions. This gives actors insight into the direction that they need to move, and the principles that need to be at the basis of a vision for the future.
- Comprehensive planning. Once the four steps describing the examples allow translation into change directions in a land-use map or landscape visual, this may very well result in a phase of chaos (Figure 1). This phase of the transition process is crucial: actors get a reality check on the guiding principles that they agreed on. This may result in a landscape that is not meeting all of their needs. An interplay between actors will then take place, resulting in the shift of needs or in the shift of guiding principles, that will lead to a landscape plan that is better fulfilling the needs, reflecting the power balance between the actors involved within the context of the landscape system. Finally, when this is settled, the landscape-based plan can be translated to action perspectives of the different types of stakeholders, resulting in a clearly defined pathway towards a new stabilized situation.

*5.2. Cherishing Abiotic Differences Enhances Sustainable and Resilient Landscapes*

The cases on water management and flood protection, in the Vallei and Veluwe and Plan Ooievaar respectively, show the importance of understanding natural processes in landscapes and knowledge of the potential that different zones in the landscape offer to enhance natural processes that deliver essential landscape services [46]. Especially regarding water management challenges, such as adaptation to climate change, the natural system offers diversity in opportunities on how measures can contribute to water management. Tailoring the measures to the opportunities the landscape offers, will make the measures more effective and efficient in delivering landscape services [47]. Also, it will result in measures that will be more sustainable, as they are in accordance with or compatible with the local natural circumstances. This will lead to a landscape where differences in natural characteristics due to hydrologic, topographic or soil differences are emphasized and considered to be an asset instead of a threat. This results in a landscape in which the natural characteristics can be highlighted instead of smoothed out, which will add value to the identity of an area within its regional context. This adds to the connection people have with the soil and landscape around them, enhancing human health and wellbeing [16,48].

*5.3. Strategies for Landscapes are More Efficient than Those for Administrative Units*

From the examples, we learned that "working with nature" needs to be done at the proper scale level, the scale level of the natural processes that deliver the desired landscape services [49–51]. In the example of the regional adaptation strategy, solutions for local floods and droughts are sought in the regional natural system, on which hydrological processes take place, connecting rural and urban areas, which is similar to other cases in different parts of the world [52–54]. The interaction between local and regional planning dimensions is crucial here [10]. In the case of the Bee Landscape, expert knowledge is used to design the required type, amount and coherence of habitat for viable populations of pollinating insects. In the case of Plan Ooievaar, the whole Dutch riverine area situated in more than four provinces and in numerous municipalities was considered as one natural system. This led to a transitional change in river management, where natural river dynamics are embraced instead of combatted [54,55]. The highly adjusted riverbed system was adapted to a more biodiverse and flood resilient system [40]. In all three cases, administrative borders were overruled, and cooperation took place on the landscape level: the level on which key natural processes take place for landscape services that were needed. Therefore, for an efficient implementation of land management strategies, it is important to take the natural limits of the system into account [56,57]. For the biosphere part of the system, the

landscape as a unit may serve as the natural planning limit, thus reducing the complexity of the whole socio-economic/bio-physical system.

### 5.4. Landscape Visions are Powerful Boundary Concepts to Define Pathways towards A Desired Future

Thinking about a desired future is always an inspiring activity. Having envisaged a specific future, pathways towards such a future can be defined [58]. A desired future may not be a strongly delineated image, also a co-created vision can be a basis for normative scenario design [30,59,60]. Landscape visions can very well serve as the vehicle for discussing future land use. As such they can be called boundary concepts in collaborative landscape governance, in analogy to the landscape services used as such by Westerink, Opdam, Van Rooij and Steingröver [29]. The image of the Bee Landscape was clearly a very inspiring local boundary concept, to bring a substantial number of parties together for a shared future of the regional landscape. Also, the idea of natural rejuvenation of the meandering river system appeared to be a powerful boundary concept to inspire many municipalities and other institutions to join forces in nature rehabilitation in the floodplains. In the Vallei and Veluwe, the "basic natural system" at a regional scale was put forward as an inspiring principle, that will enable the scientific and governance community to identify adequate local pathways towards a desired future.

### 5.5. Co-Creation Is Essential to Safeguard Adaptive, Circular and Biodiverse Landscapes

Successful landscape-based planning is characterized by working on the required scale level, choosing the required dimension of the planning area, always embedded in the next higher level. The use of landscape services emphasizes the potential of the landscape as a means to realize a desired future. All sectors present in an area are involved in the planning, which is a basic character of co-creation, where public and knowledge institutions collaborate not only with private bodies but also with civil society to innovate services and products [61,62]. Therefore, as all sectors have their own perspective on issues, language and context, it appears that a boundary concept helps to enable the conversation and overcome differences between stakeholders. In addition, the cases show that a time horizon in the far future helps to overcome discussions on current problems and enables focus on possibilities. Recently, a vision on a natural future for the Netherlands in 2120 appeared, showing how the Netherlands would look like when taking the landscape as a guiding principle for adaptation to climate change in a biodiverse and circular environment. Stakeholders and authorities of all sorts found this very interesting and mind-shifting, giving way to further, out of the box discussions on the issues that are ahead [18,63].

### 5.6. Landscape-Based Approaches Enhance Future-Proof Land-Use Transitions

Transitions in the context of spatial planning can only be recognized as such after completion. The evaluation of the cases shows that the landscape-based planning approach stands for land-use transitions based on a landscape-based spatial development, that is, an approach to spatial processes that takes into account all the relationships in the landscape: both the physical landscape with its layers and functions and the socio-ecological landscape with its different scales and actors. This provides a basis for a transition framework towards sustainable land use in the long term, which in turn gives direction to necessary short-term actions in land restoration, nature rehabilitation and innovative forms of land use. Although change is often difficult to bring about, the landscape itself shows the traces of constant change in a positive way—its particular character does not need to suffer from change. Normative futures as boundary concepts can help—instead of building our present on our past—to learn and build our present on our future, on what is possible, instead of merely on what has gone before [4,8,30].

## 6. Conclusions

As illustrated with the three examples, the landscape-based planning approach enhances a development towards a future land system resilient to external pressures—at

least the foreseeable ones. The concepts of nature-based solutions and transition theory are fundamentally combined in this approach, where co-created normative future visions serve as boundary concepts in the regional spatial planning debate. A shared long-term vision of what our future landscape should look like is a crucial source of inspiration for a coherent design approach to solve today's spatial planning problems, such as climate adaptation, biodiversity enhancement and circular resource management. The landscape-based approach principally uses the natural characteristics of the landscape system as an opportunity instead of a limitation. It gives direction to the technical-economic preconditions for sustainable landscape development, such as drainage standards and environmental quality. Rather than as an object in itself, the landscape is considered as a comprehensive principle, to which all spatial processes are inherently related. Local planning can only be adequate when logically embedded in a regional perspective. The main recommendation for future research is therefore that solutions to regional planning problems should be studied that go beyond the traditional nature-based solutions, by emphasizing the spatial dimension, the specific time horizon considered and the interaction of all sectoral considerations of the urban and rural landscape. Special attention should be paid to the adequateness of social involvement and participation, which is to be defined for each case differently, and which could easily play a disturbing role. Also, the dominance of strong short-term economic functions such as transport, housing, energy provision is currently often triggering trade-offs, especially when the shared long-term vision is not accompanied by (inter)national instruments to guide sustainable developments at lower spatial scales [64,65]. At the end of the day, however, the landscape-based planning approach should allow professionals, researchers, stakeholders and citizens alike, to participate in the transition to a forward-looking normative design. Working towards such a future, pathways can be defined towards a shared vision, observing the boundaries of a safe operating space.

**Author Contributions:** Conceptualization, S.v.R., W.T., O.R. and B.P.; methodology, S.v.R., W.T., O.R. and S.K.; writing—original draft preparation, S.v.R., B.P. and M.S.; writing—review and editing, B.P. and M.S.; visualization, S.K., W.T. and S.v.R.; supervision, S.v.R.; project administration, O.R.; funding acquisition, S.v.R. and O.R. All authors have read and agreed to the published version of the manuscript.

**Funding:** This research was funded by the Netherlands Ministry of Agriculture, Nature and Food Quality, grant number KB-34A-007-008 1-2C-6 (Circular & Climate Neutral), and KB-36-005-006/008 (Nature-inclusive Transitions).

**Conflicts of Interest:** The authors declare no conflict of interest. The funders had no role in the design of the study; in the collection, analyses or interpretation of data; in the writing of the manuscript or in the decision to publish the results.

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
