# Peer review of "Landscape-Based Visions as Powerful Boundary Objects in Spatial Planning: Lessons from Three Dutch Projects"

_land, doi:10.3390/land10010016_

Round 1
Reviewer 1 Report
I enjoyed reading this paper. The contribution is clear, and the methods are adequately described. The only suggestion I would make is to tie the cases more directly into the literature, as way of explaining within each case what was being referenced. The discussion does this well, but I found that as I read the cases I wanted a bit more guidance as to where they were going. This may be a reflection of me as a reader more than the cases themselves. I think the issues addressed will be of interest to many readers.
Reviewer 2 Report
The research presented by author has been devoted to an important issue that is landscape-based planning approach to regional spatial policy. It is important, comprehensive and new. Local planning logically fixed in regional perspective has high added value. Relationship between local and regional planning will be interesting for scientists, PhD students and many other readers. The article meets high requirements of the journal, after minor revisions needs to be supplemented by the mentioned remarks and is recommended for publishing.
- It is advisable to explain more detailed why exactly such three examples (Vallei & Veluwe, Bee landscape and Plan Ooievaar) have been selected for investigation.
- The discussion provides too little information about interaction between local and regional planning and their reinforcement.
- It is advisable to explain more detailed an essence of use of natural characteristics of the landscape system as an opportunity, as well as technical-economic preconditions for sustainable landscape development. If there are data, it is advisable to add them.
- It is important to present the recommendations for future studies.
Reviewer 3 Report
This paper presents an interesting landscape-based planning approach that, combining the concept of nature-based solutions and transition theory, is aimed at regional spatial policies allowing for a community-based transition. In particular, three examples from the Netherlands are presented to show how this approach can be used as a guideline for land transition, by considering landscape as a vehicle for spatial planning.
The paper is well structured and its aim is clear. In order to improve the quality of this paper I would recommend the following minor revisions:
- THE X-CURVE TRANSITION CROSS (Figure 1): I think that a correct explanation of this conceptual model is fundamental to understand transitions in landscape adaptation processes, generally and in relation to the three examples. For this reasons, I suggest to more deeply describe this transition process in each of its phases.
- PAR.4: the three examples are clearly illustrated in each of the 4 guidance steps. I would appreciated also a graphical representation obtained from the conceptual model previously described (Figure 1), in order to highlight the main differences between the different landscape-based approaches. Although in paragraph 5.1 the lessons learnt from the examples are illustrated, as well as the main issues are effectively compared and synthesized, I think that a more detailed description of each landscape-based approach would improve the description of the assumptions on which the discussion (par.5) is based.
- CONCLUSION: The Authors do not properly focus the limitations of this study: I suggest to have a more critical approach and to highlight those aspects that could be developed and improved in future researches.
Reviewer 4 Report
This paper aims to introduce the concept of a landscape-based planning approach. It is using three examples from the Netherlands.
Questions to be answered are how a landscape-based approach adds to nature-based solutions, how the abiotic landscape can be considered in terms of opportunities, how future visions can support sound transitions, and how local and regional planning can reinforce each other.
The concept is original and can be useful, as shown in the Netherlands' case that urban and rural are mixed. Meanwhile, how to generalize for any country?
In line 143, to what part of the world is referring? "the concept of landscape services in this part of the world is more appropriate than the concept of ecosystem services." Probably only the Netherlands world, thus the research is of local interest and should be published in a local journal. To avoid this misunderstanding it is needed to generalize the concept. Thus I recommend avoiding localizing the concept and limiting to this part of the world.
In chapter 3, in general, and more specifically in lines 137, 142, 149, the authors try to convince that using “landscape-based planning” rather than “landscape-based solutions” is better. In reality, “landscape-based planning” is a “nature-based solution.” Chapter 3 needs some references from recent EU projects about nature-based solutions upscaled to the landscape level instead of local demonstrations of individual NBS. In this chapter, it should be given a clear message to the first question of the manuscript: "how a landscape-based approach adds to nature-based solutions."
The term co-creation appears 5 times in the manuscript, and it is not clear what is this process. In chapter 5.5 is needed to define what it is about. It is not about a multidisciplinary team creating something and demonstrating it to stakeholders, as in the last paragraph of this chapter. Please mention co-design and co-creation in urban living labs aproach.
In chapter 5.1, line 328, please add a citation about adaptive planning.
The conclusions are informative and to the point.
The abstract is like an extended introduction with a conclusion. It is not mentioning the aim of the paper in a clear phrase. It is not mentioning the case studies method from the Netherlands and anything about the lessons learned from them.
In general, the paper is interesting to an international audience of both land planners and nature-based solutions specialists. Thus, I recommend to published after minor improvements given above.
